# LIM Kinases in Osteosarcoma Development

**DOI:** 10.3390/cells10123542

**Published:** 2021-12-15

**Authors:** Régis Brion, Laura Regnier, Mathilde Mullard, Jérome Amiaud, Françoise Rédini, Franck Verrecchia

**Affiliations:** 1INSERM UMR1238, PHY-OS, Bone Sarcomas and Remodeling of Calcified Tissues, Nantes University, 44000 Nantes, France; regis.brion@univ-nantes.fr (R.B.); laura.regnier@univ-nantes.fr (L.R.); mathilde.mullard@univ-nantes.fr (M.M.); jerome.amiaud@univ-nantes.fr (J.A.); francoise.redini@univ-nantes.fr (F.R.); 2Centre Hospitalier Universitaire, Université de Nantes, 44000 Nantes, France

**Keywords:** LIMK, osteosarcoma, therapeutic target

## Abstract

Tumorigenesis is a long-term and multistage process that often leads to the formation of metastases. During this pathological course, two major events appear to be crucial: primary tumour growth and metastatic expansion. In this context, despite research and clinical advances during the past decades, bone cancers remain a leading cause of death worldwide among paediatric cancer patients. Osteosarcomas are the most common malignant bone tumours in children and adolescents. Notwithstanding advances in therapeutic treatments, many patients succumb to these diseases. In particular, less than 30% of patients who demonstrate metastases at diagnosis or are poor responders to chemotherapy survive 5 years after initial diagnosis. LIM kinases (LIMKs), comprising LIMK1 and LIMK2, are common downstream effectors of several signalization pathways, and function as a signalling node that controls cytoskeleton dynamics through the phosphorylation of the cofilin family proteins. In recent decades, several reports have indicated that the functions of LIMKs are mainly implicated in the regulation of actin microfilament and the control of microtubule dynamics. Previous studies have thus identified LIMKs as cancer-promoting regulators in multiple organ cancers, such as breast cancer or prostate cancer. This review updates the current understanding of LIMK involvement in osteosarcoma progression.

## 1. Introduction

Despite advances in the therapeutic treatment of osteosarcoma (OS), the leading malignant bone tumour in children and adolescents, many patients die from this disease. Whereas the 5-year survival rate is greater than 78% in the absence of metastases at diagnosis or for good responders, this survival rate drops to 25% for metastatic or recurrent OS [1,2,3,4]. These survival rates, which have shown little or no improvement in recent decades, underscore the urgent need to develop new therapeutic strategies. In this context, LIM kinases (LIMKs), which play a major role in controlling actin microfilament dynamics and microtubule (MT) dynamics, have been described as regulators of the development of multiple cancers.

## 2. LIMKs and Cancer

### 2.1. The LIMK Family

The LIMK family consists of ubiquitous proteins composed of two members: LIM kinase 1 (LIMK1) and LIM kinase 2 (LIMK2), encoded in humans by two different genes located, respectively, at 7q11.23 and 22q12.2. The pre-messenger RNAs of LIMK1 and LIMK2 undergo alternative splicing processes, resulting in the synthesis of different proteins from a single genomic sequence. The gene encoding LIMK1 consists of 39,499 base pairs. Two splice variants of LIMK1 are known to date: LIMK1-1 and LIMK1-2. LIMK1-1 is thought to encode the full-length protein, and LIMK1-2 lacks the kinase domain [5,6,7]. The gene coding for LIMK2 consists of 68,671 base pairs. Three splice variants were identified: LIMK2a and LIMK2b, which code for a smaller protein, and LIMK2-1, which generates a protein corresponding to the N-terminal variant 1 and C-terminal variant 2ab. In addition, tLIMK2, a testis-specific LIMK2 variant, has been identified [8,9,10].

#### 2.1.1. Protein Structure

The canonical sequences of each of the LIMKs have two LIM domains at the N terminus, named for the acronym of three transcription factors (Linl1, Isl1, and Mec-3), a PDZ domain (postsynaptic density protein-95, Drosophila dise large tumour suppressor, and Zonula occludens-1 protein), a serine- and proline-rich S/P domain, and a catalytic kinase domain in the C-terminus (Figure 1) [5,6]. The LIM domains contain two zinc finger motifs that inhibit the activity of the kinase domain by interacting with it [11,12] and contributing to the formation of homodimers and heterodimers [13]. The PDZ domain, which, like the LIM domains, allows protein-protein interactions, is also important for LIMK trafficking between the cytoplasm and the nucleus because it contains two leucine-rich nuclear export signals (NESs) [14,15,16]. One functional nuclear localisation signal (NLS) has been identified in the kinase domain of both LIMK1 and 2. The two sequences that differ between the two LIMKs, the LIMK1′s eight amino-acid sequences lead to nuclear localisation, whereas the LIMK2′s 13 amino-acid sequences trigger a nuclear/nucleolar localisation. A putative NLS sequence between the PDZ and the kinase domain has also been identified. However, the NLS sequence does not seem to be functional. Two additional NLSs have been identified in the LIMK structure. The first, identified on LIMK1 and LIMK2, is located between the PDZ domain and the catalytic domain of LIMKs. The second was identified only in the catalytic domain of LIMK2 [16]. The kinase domain of LIMKs allows the phosphorylation of substrates, including cofilin, mainly through serine/threonine kinase activity [12,13,14]. Finally, the LIMK2-1 isoform-specific PP1i domain appears to be unable to phosphorylate cofilin directly but seems to stabilize its phosphorylation through the isoform specific PP1i domain inhibition of phosphatase 1. The homology between the two proteins, LIMK1 and LIMK2, is estimated to be 50%, with 70% homology in the catalytic domain, 50% homology in the LIM domains, and 46% in the PDZ domain [13].

#### 2.1.2. LIMKs and Actin Cytoskeleton or MT Rearrangement

LIMKs are primarily known to phosphorylate the ADF/cofilin family of proteins comprising mainly three members: cofilin 1, cofilin 2, and destrin (also known as actin depolymerisation factor, ADF) [17]. The cofilin family plays an essential role in actin dynamics by promoting the severing and depolymerisation of actin filaments [18,19,20,21,22]. LIMKs regulate cofilin activity by inactivating it through phosphorylation of a serine residue at position 3 (Ser3) (Figure 1) [23,24,25]. The activation of cofilin by dephosphorylation is mainly achieved by phosphatases, such as slingshots (SSH) [26], chronophin (CIN) [27], and phosphatases 1 and 2A (PP1/2A) [28]. In addition, SHH is able to exert its phosphatase activity directly on LIMK1 [29]. The alternation of phosphorylation and dephosphorylation phases of cofilin thus plays a fundamental role in the reorganisation of the actin cytoskeleton and in many cellular functions [7,19,30].

In addition to regulating the dynamics of the actin cytoskeleton, LIMKs appear to contribute to MT rearrangement [31]. The molecular mechanisms behind these processes remain poorly defined and controversial. Initial studies have shown that LIMK1 and LIMK2 phosphorylate TPPP (Tubulin polymerization-promoting protein) [32]. Phosphorylation of TPPP is thought to inhibit its ability to promote tubulin polymerisation [32,33]. In addition, LIMK2 is thought to regulate the organisation of astral MTs and the orientation of the mitotic spindle orientation during cell division in HeLa cells through its action on TPPP [33]. However, other data have shown that LIMKs do not interact with TPPP, which is phosphorylated directly by the ROCK protein [34]. LIMKs could regulate MTs by interacting directly with tubulin [35].

**Figure 1 cells-10-03542-f001:**
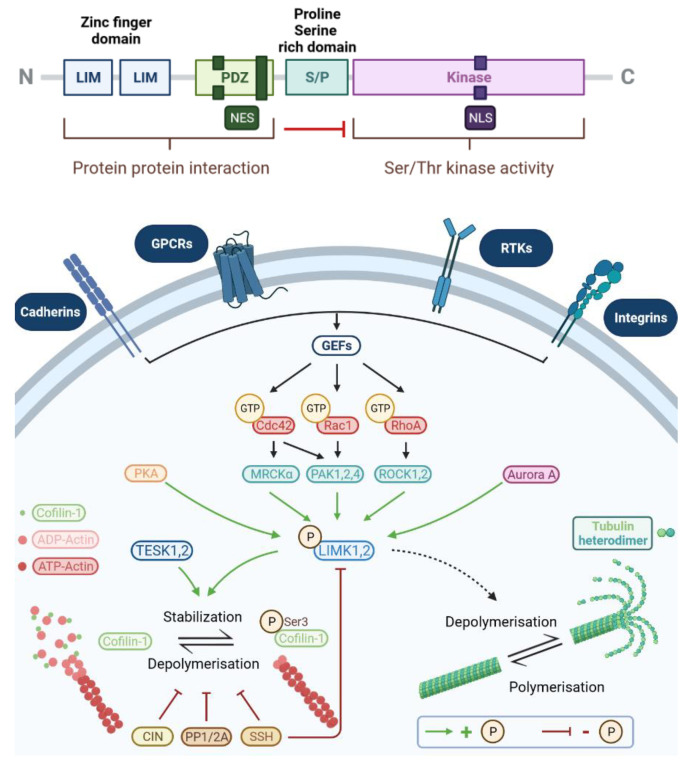
The LIMK family: (**Upper panel**) Protein organization of LIM kinases. Blue represents the LIM domains, green represents the PDZ domain, blue/green represents the S/P domain, and pink represents the kinase. (**Lower panel**) Activation of LIMKs. LIMKs are phosphorylated by the RhoA/ROCK1,2, Rac1/PAK1,2,4, Cdc42/MRCKα, or Cdc42/PAK1,2,4 pathways upon stimuli. LIMK1/2 regulate cofilin activity by its phosphorylation. Cofilin binds preferentially to actin-ADP, stimulates actin cleavage and depolymerization, and promotes actin filament turnover. Cofilin is inactivated by LIMK through the phosphorylation of a serine residue at position 3 (Ser3). The activation of cofilin by dephosphorylation is mainly achieved by phosphatases such as slingshots (SSH), chronophin (CIN), and phosphatases 1 and 2A (PP1/2A). LIMKs also contribute to microtubules (MT) rearrangement.

#### 2.1.3. Activation of LIMKs

The activity of LIMKs is mainly regulated by small G proteins of the Rho-GTPase family [36], such as Rho, Rac, and Cdc42 (Figure 1) [5,37,38,39,40,41]. These small G proteins activate, in particular, the PAK (p21 protein-activated kinase), MRCK (myotonic dystrophy kinase-related Cdc42-binding kinase), and ROCK (Rho-associated protein kinase), which activates LIMKs. Briefly, PAK proteins preferentially phosphorylate LIMK1 downstream of Rac1 and Cdc42 [40,42,43,44]. The Rho/ROCK1 and 2 pathway preferentially activates LIMK2 [37,38,39,45], although when overactivated, ROCK is also capable of phosphorylating LIMK1 [45,46]. Both ROCK and PAK can directly phosphorylate LIMKs at Thr508/Thr505. Moreover, ROCK and PAK can phosphorylate LIMK1 at Ser323/Ser596 and, thus, its kinase activity [47]. The MRCKα (myotonic dystrophy kinase-related Cdc-42-binding kinase α) protein activates both LIMK1 and LIMK2 downstream of Cdc42 [38]. Other protein kinases can phosphorylate and activate LIMKs. For example, protein kinase A (PKA) phosphorylates LIMK1 [47] on two serine residues at positions 96 and 323, and the MAPK protein kinases phosphorylate LIMK1 [48] on a serine residue at position 323, while the Aurora-A phosphorylates LIMK1 [49] on a serine residue at position 307 and a threonine residue at position 508 and LIMK2 [50] on a serine residue at position 283, a threonine residue at position 494, and a threonine residue at position 508. Moreover, the homodimerization of LIMK1 promotes trans-phosphorylation of the kinase domain, promoting the stability of the kinase and increasing its activity [7].

### 2.2. LIMKs and Cancer

The role of LIMK/cofilin signalling has been studied in several types of cancers and cancer lines, such as breast cancer [51,52,53], colon cancers [54,55], prostate cancers [56,57], ovarian cancers [58], pancreatic cancers [59], and lung cancers [60,61]. In this context, the role of LIMKs in cancer development is closely associated with the ability of LIMKs to regulate the architecture of the actin cytoskeleton or the rearrangement of MTs and, thus, key functions of tumour development such as cell proliferation or migration [62,63]. Thus, among the many activators of LIMK, members of the Rho family of small GTPases (Rho, Rac, and Cdc42) and their effectors (ROCK, PAK, and MK2) are implicated in the progression of various human cancers to invasive and metastatic stages [62].

#### 2.2.1. LIMKs and Cell Cycle

Both actin microfilament dynamics and MT rearrangement play a crucial role in cell cycle control. In this context, many studies have demonstrated the role of LIMK1 and LIMK2 in the regulation of various stages of mitosis, such as mitotic spindle organisation, chromosome segregation, and cytokinesis [64]. The localisation of LIMKs varies according to the different stages of cell division. LIMK1 is expressed in the cytoplasm of cells during interphase, at the poles of the mitotic spindle from prophase to metaphase [64], and at the contractile ring during cytokinesis [65,66]. For this reason, LIMK1 is thought to be involved in mitotic spindle orientation and cytokinesis. For example, the extinction of LIMK1 expression or inactivation leads to mitotic spindle disorientation. In contrast, increasing the level of cofilin phosphorylation through the overexpression of LIMK1 increases the number of multinucleated cells [66,67]. LIMK2 is located at the mitotic spindles during the metaphase and anaphase [64]. LIMK2 is thought to participate in cell cycle regulation by acting on both cyclins and cofilin. For example, the formation of the D1-CDK4 complex, required for S-phase transition, leads to the overexpression of LIMK2 [68], while translocation of LIMK2 into the nucleus decreases the expression of cyclin D1 [69,70]. Phosphorylation of LIMK2 by ROCK promotes the expression of cyclin A, which is implicated in the transition from S to G2 phase [70].

#### 2.2.2. LIMKs and Cell Migration

Cell migration is finely regulated by the reorganization of the actin cytoskeleton [71,72]. During migration, the brush-like ends of actin filaments are oriented toward the plasma membrane at the filopodia and lamellipodia. The elongation of these filaments pushes the cell edge forward [73], while the stress fibres, composed of actin and myosin, provide the contractile forces necessary for migration [74]. These different stages of cell migration are finely regulated by LIMK1 and LIMK2. Briefly, overexpression of LIMK1 leads to aberrant accumulation of actin filaments and, thus, inhibits cell motility [75]. Conversely, the silencing of LIMK1 expression blocks lamellipodia formation and, thus, cell migration [76,77]. Phosphorylation, and the corresponding inactivation of cofilin by LIMK1, promote lamellipodia formation and extension by stabilizing actin filaments, but also inhibit lamellipodia extension by decreasing G-actin levels [78]. Notably, LIMK1 and LIMK2 may contribute distinctly to these cell migration processes, since these kinases are activated by different pathways. Indeed, Rac1 and Cdc42 are involved in the formation of lamellipodia and filopodia and preferentially activate LIMK1, whereas RhoA is mainly associated with stress fibre formation and preferentially activates LIMK2 [79].

## 3. LIMKs and Primary Bone Tumour

### 3.1. LIMKs and Bone Remodeling

Bone remodelling is critical to the development of primary bone tumours such as OS. A vicious circle is established between the tumour cells and bone cells during OS development. Cancer cells produce soluble factors that can directly or indirectly modulate bone degradation. Indeed, OS cells produce osteolytic cytokines such as RANKL, which directly stimulate the differentiation and activation of osteoclasts. OS cells can also produce cytokines stimulating the activity of osteoblasts and, thus, the production by osteoblasts of osteolytic cytokines, such as RANKL. These cytokines, in turn, stimulate the differentiation and activity of osteoclasts. The activation of osteoclasts leads to a stimulation of bone osteolysis and to the subsequent release of protumoural factors trapped in the bone matrix, which then stimulate tumour development [80].

#### 3.1.1. Brief Overview of Bone Remodelling

Bone is a specialized, living, and constantly evolving connective tissue that forms, together with multiple types of tissue, the structural elements of bones. Therefore, bone tissue is a site of continuous remodelling involving mainly two cell lineages: the osteoblastic mesenchymal lineage (osteoblasts) and the osteoclastic hematopoietic lineage (osteoclasts). Schematically, osteoblasts and osteoclasts direct bone formation and resorption, respectively [81].

(a)Osteoblastogenesis

Osteoblastic progenitors include mesenchymal stem cells (MSCs) (Figure 2) [82,83,84] and present mainly in the bone marrow. These progenitors undergo a series of differentiation steps that lead to the formation of mature osteoblasts. As they differentiate, osteoblasts acquire the ability to form bone tissue, and are responsible for the synthesis and deposition of the organic components of the bone matrix and for orchestrating its mineralization [85,86,87]. These differentiation processes are controlled by various cytokines that stimulate the activity of transcription factors [88]. Among these transcription factors, Runx2 and OSX centrally regulate osteoblastogenesis [89,90,91,92,93,94]. Runx2 is a transcription factor from the RUNX family, consisting of RUNX1, RUNX2, and RUNX3, which can regulate the transcription of many osteoblastogenesis markers, such as ALP (alkalin phosphatase), OCN (osteocalcin), OPN (osteopontin), osteonectin (ONN), BSP (bone sialoprotein), and COL1A1 (type I collagen α1 chain) [88,95,96]. This is achieved mainly by binding to a consensus site along the proximal promoters of the genes as identified on the promoters of type I collagen α1 chain, bone sialoprotein, osteocalcin, or osteopontin [97]. During the in vitro differentiation of MSCs into osteoblasts, RUNX2 is very weakly expressed in MSCs prior to their commitment to the osteoblastic lineage and increases throughout pre-osteoblast proliferation. The level of RUNX2 peaks at the immature osteoblast stage and then decreases during osteoblast maturation [98]. In vivo, Runx2 is expressed in all mesenchymal condensations before osteoblast differentiation during early murine skeleton development at E10.5–E12.5 [99], and the deletion of Runx2 in mice causes defects in endochondral and intramembranous bone formation and, ultimately, a failure of bone formation [100,101]. In humans, RUNX2 haplo-insufficiency causes cleidocranial dysplasia, characterized by short stature, delayed fontanel closure, prominent forehead, drooping shoulders, and abnormal dental development [102,103]. Furthermore, RUNX2 stimulates the expression of osterix (OSX), another major transcription factor in the differentiation of MSCs into osteoblasts. OSX, a transcription factor containing three C2H2-type zinc fingers, stimulates the expression of the markers ALP, OCN, OPN, ONN, BSP, and COL1A1 mainly by binding to GC-rich regions on the proximal promoter of these genes. Expression of OSX in osteoblast precursors induces the differentiation of these cells into mature and functional osteoblasts [104]. In this context, OSX-deficient mice have also shown defects in bone formation due to the complete loss of osteoblasts [105]. In humans, a homozygous mutation in the OSX gene has been identified in a patient with a moderate form of osteogenesis imperfecta, presenting with bone fractures, mild bone deformities, and delayed tooth eruption [106].

(b)Osteoclastogenesis

Osteoclasts are large, multinucleated cells specialized in bone resorption by proteolytic degradation and acid decalcification of the bone matrix. These cells are derived from osteoclast precursors of hematopoietic origin [107] that differentiate into osteoclasts in response to the master osteoclastogenic cytokine, RANKL (receptor activator of nuclear factor-κB ligand), and M-CSF (macrophage colony-stimulating factor), which are produced by osteoprogenitor mesenchymal cells and osteoblasts [80,108]. M-CSF binds to cFMS receptors on osteoclast precursors and regulates their survival and proliferation [109]. RANKL regulates fusion and maturation osteoclast precursors [109]. Schematically, RANKL binds to the RANK receptors present on the membrane of osteoclast precursors [110] and recruits TNF receptor-associated factor 6 (TRAF6) to induce various signalling cascades, including canonical and non-canonical NF-kB pathways and mitogen-activated kinase (MAPK) pathways [111,112,113]. In the early stage of osteoclastogenesis, RANKL activates AP-1 transcription factor comprising c-Fos to drive osteoclast differentiation [114,115]. In the late stage of osteoclastogenesis, c-Fos cooperates with the nuclear factor of activated T-cells cytoplasmic 1 (NFATc1) to regulate osteoclast fusion [109]. In contrast, OPG (osteoprotegerin), produced by osteoblasts, inhibits bone resorption by binding to RANK, which prevents RANK/RANKL interaction and osteoclastogenesis [116]. Thus, the RANKL/RANK/OPG axis is a primary pathway to mediate osteoclast differentiation.

#### 3.1.2. LIMKs and Bone Remodelling

The literature on the role of LIMK-kinases in skeletal development is sparse. However, Kawano et al., examining the skeletal phenotype of LIMK1 knock out mice, showed a decrease in bone mass in the absence of LIMK1. Indeed, H&E staining and micro-CT analyses of the femurs of LIMK1^−/−^ and LIMK1^+/−^ mice have indicated that the trabecular bone mass is reduced in LIMK1^−/−^ mice, with significantly reduced values for the trabecular bone fraction, trabecular number and thickness, and increased intrabecular spacing. In vitro analyses have shown impaired osteoblastic and osteoclastic functions in the absence of LIMK1. Regarding osteoblastogenesis, fewer osteoblast progenitors have been observed in the marrow of LIMK1^−/−^ animals. Furthermore, the ability of osteoblasts cultured from LIMK1^−/−^ mice to mineralise is reduced, suggesting that not only is the number of osteoblasts reduced but also their ability to differentiate. Regarding osteoclastogenesis, the ability of osteoblasts cultured from LIMK1^−/−^ mice seem to have their resorptive capacity considerably diminished. This work shows that LIMK1 plays a crucial role in bone remodelling and skeletal development, as it is necessary for normal osteoblast differentiation and bone resorption by osteoclasts [117]. In the context of the differentiation of MSCs into osteoblasts, Chen and Coll have shown that stabilisation of actin filaments stimulates the differentiation of MSCs into osteoblast cells in vitro and heterotopic bone formation in vivo. At the molecular level, the inhibition of cofilin phosphorylation by LIMK1 inhibition decreases the differentiation of MSCs into osteoblasts, with a decrease in the expression of a differentiation marker, such as alkaline phosphatase, and a decrease in the ability of the cells to mineralize [118]. In this context, Liu and Coll have shown that BMP2 accelerates MSC migration and recruitment via the CDC42/PAK1/LIMK1 pathway both in vivo and in vitro [119]. The role of LIMK1 in MSC migration has been confirmed by Chen and Coll, who showed that KIAA1199, a factor secreted by MSCs during their differentiation into osteoblasts [120], regulates MSC migration by regulating the actin depolymerisation factors Cofilin1, LIMK1, and Destrin [121].

The role of LIMKs in the reorganisation of the actin cytoskeleton in osteoblasts has also been demonstrated for LIMK2. Indeed, Fu and Coll showed that LIMK2 in murine osteoblast lines crucially supports the organisation of the actin cytoskeleton in osteoblasts in response to fluid shear stress [122]. These results have been confirmed by Yang et al., who showed that LIMK2 silencing inhibited the fluid shear stress-induced reorganisation of the actin cytoskeleton of primary osteoblasts and demonstrated that the mechanosensitivity of osteoblasts in response to this stress was enhanced [123].

Thus, it appears that LIMK-Kinases (LIMK1 and LIMK2) participate essentially in the control of bone remodelling by controlling the reorganisation of the actin cytoskeleton.

### 3.2. LIMKs and Pediatric Bone Tumors

#### 3.2.1. Osteosarcoma (OS)

OS (Table 1), first described in 1806 by Dr. Alexis Boyer, is, according to the World Health Organization’s definition, a malignant tumour characterized by the production of bone or osteoid substance by the tumour cells [124]. According to this definition, even minimal production of bone is sufficient to call it OS. OS actually presents a wide variety of lesions distinct in their clinical and radiographic presentation, microscopic appearance, and evolution, leading to the definition of this pathology as a highly heterogeneous tumour [125]. Three main histological subtypes of OS have been defined: osteoblastic OS (50% of cases), chondroblastic OS (25% of cases), and fibroblastic OS (25% of cases).

Childhood cancers are rare, accounting for approximately 1% of all cancers [126,127]. Among these tumours, OS is the most common primary bone tumour in children, adolescents, and young adults [128,129]. Worldwide, the incidence is 3.4 per million per year and 4.4 per million for ages 0–24 [130]. As a function of age, OS is characterized by a bimodal distribution with a first peak between 0 and 24 years, and a second between 75 and 79 years [131]. The first peak, corresponding to puberty, suggests a close relationship between the onset and development of the disease and bone growth or remodelling. The tumour most commonly develops in the femoral diaphysis (40% of cases), tibia (20% of cases), and humerus (10% of cases). The tumour can also develop in flat bones, such as the pelvis (8% of cases) [3].

**Table 1 cells-10-03542-t001:** Main features of osteosarcoma.

Osteosarcoma	
Paediatric bone tumor ranking	1st
Incidence (worldwide)	3.4 per million per year
Age of patients	18
Preferential localization	Long bones
Cellular origin	Mesenchymal cells or osteoblast
Histology	Osteoblastic/chondroblastic/fibroblastic
Main metastases localization	lungs
Treatment	Chemotherapies/chirurgical resection/chemotherapies
5-year survival	70/75% (localised form) or 20/25% (metastatic form)

Two hypotheses address the question of the cell of origin of OS. The first suggests that the cells of origin are mesenchymal cancer stem cells that can differentiate into different cell types, which would contribute to tumour heterogeneity. The second hypothesis suggests that a single cell (poorly differentiated osteoblast) undergoes an accumulation of genetic mutations that allow the acquisition of stem cell properties to result in tumour heterogeneity [124,132,133]. Regardless of the hypothesis, the expression of markers such as RUNX2 (Runt-related transcription factor 2), ALP (alkaline phosphatase), OCN (osteocalcin), and BSP (bone sialoprotein) by OS cells suggests that this tumour originates from the deregulation of the differentiation program of cancerous or non-cancerous MSCs [133,134].

OS cells are characterized by a complex karyotype, a large number of chromosomal breaks, and a high frequency of genetic alterations [135]. No specific gene has been identified, but several genes have been found to be involved in the development of OS. For example, whole genome sequencing analyses have shown that the tumour suppressor gene p53 or Rb1 (retinoblastoma 1) is found in these areas of chromosomal instability, and a defect in p53 or Rb1 is identified in most OS [132,135,136,137]. A deletion of the WWOX gene, which plays a suppressive role in OS, is observed in 30% of OS [138]. Some oncogenes, such as c-Myc, are overexpressed in metastatic OS and correlated with poor survival [139]. Although the exact origin of OS remains unknown, various pathologies are closely associated with the development of OS [140]. For example, Li-Fraumeni syndrome, which is related to a germline mutation in the p53 gene, leads to the development of multiple tumours such as OS [141,142]. A mutation in both alleles of the Rb1 gene, which leads to the development of retinoblastoma, is also a predisposition to OS. Patients affected by this mutation have a 500-fold increased risk of developing OS [3,131]. This alteration is found in 70% of OS, and it is correlated with a poor prognostic marker [143]. Rothmund-Thomson syndrome, linked to a mutation on chromosome 8, is also characterized by an increased risk of OS [144]. Paget’s disease, associated with abnormalities of bone architecture and bone remodelling, also increases the risk of developing OS [145].

The development of OS is closely associated with the activation of various signalling pathways. Recently, we showed that the Hippo/YAP pathway is involved in the regulation of OS development both by promoting primary tumour growth through the association of YAP with the transcription factor TEAD and by promoting metastatic development through the association of YAP with the transcription factor Smad3 [146,147,148]. Interestingly, cytoplasmic YAP is able to regulates the expression of the Rho GTPase-activating protein known as the regulator of the RhoA/LIMK/cofilin pathway [149,150], suggesting a potential role of the RhoA/LIMK/cofilin cascade in OS development.

One of the main problems for patients as OS progresses is the development of mainly pulmonary metastases in 20% of patients. Before the 1970s, survival rates for patients with OS were very low, even for patients without metastases at diagnosis. The only treatment was amputation. However, 6–12 months after this surgery, lung metastases developed. Today, more than 80% of patients retain their limbs, and better long-term survival is observed with a combination of surgical resection of the tumour and chemotherapy using four classical chemotherapy agents: cisplatin, doxorubicin, methotrexate, and ifosfamide [3,151]. Whereas this conventional treatment has significantly improved patient survival, especially for high-grade OS, no truly effective treatment is currently available for metastatic OS or for patients with poor response to chemotherapy [152]. In this context, many innovative therapies are under study to improve survival in patients with aggressive or recurrent OS. More than 500 studies are ongoing worldwide: Available online: https://clinicaltrials.gov (accessed on 14 December 2021).

#### 3.2.2. LIMKs and OS

Concerning LIMK1, several studies have shown an increase in LIMK1 expression in OS tumour tissues or OS cell lines (Figure 3). Indeed, in a recent study, Li and Coll examined 56 patients and determined an increase in LIMK1 protein levels in tumour tissues, as compared to the expression measured in adjacent healthy tissues [153]. These results confirm those obtained by Yang and Coll on a smaller number of six patients [154]. The increase in LIMK1 expression was also observed in a comparison of expression in healthy osteoblastic cells and several OS lines, including MG63, U2OS, OS732, Saos2, and vincristine-resistant MG63 cells [153,154,155]. Interestingly, the expression of LIMK1 and its substrate, Cofilin-1, in OS is associated with clinical stage, distant metastasis, and tumour grade. On the other hand, no correlation has been observed between the size or location of the tumour, LIMK1 and Cofilin-1 expression, and age or sex of the patients [153].

Regarding the role of LIMK1 in the regulation of key functions associated with tumor development, converging studies demonstrated the crucial implication of LIMK1 in OS cell proliferation. Indeed, the overexpression of LIMK1 stimulates the proliferation of MG63 cells [153]. In this context, Zhang and Coll showed that LIMK1 overexpression is implicated in the control of OS cell proliferation by insulin. Indeed, the extinction of LIMK1 blocks the ability of insulin to stimulate the proliferation of OS cells [156].

LIMK1 also seems to control OS cell migration. Indeed, the silencing of LIMK1 in vincristine-resistant MG63 cells decreases their ability to migrate [155]. Moreover, the overexpression of LIMK1 in MG63 cells stimulates the cells’ ability to migrate and invade [153]. In this study, the authors further demonstrated that PAK4, which is known to regulate cytoskeletal reorganization by LIMK1/cofilin phosphorylation [42], triggers cell migration in OS cells through LIMK1/cofilin involvement. Using an OS mice model with subcutaneous injection of OS cells overexpressing LIMK1 or underexpressing PAK4, the authors showed the crucial function of the PAK4/LIMK1 cascade in the development of OS in vivo [153].

In the context of the role of LIMK1 in OS tumour progression, some studies have shown that different drugs act on OS cells via LIMK1. Yoshizawa and Coll demonstrated that 6-Hydroxythiobinupharidine isolated from *Nuphar pumilum* inhibits migration of LM8 OS cells by decreasing the expression of LIMK 1 [157]. More recently, Zhang and Coll showed that fucoidan from the sea cucumber *Cucumaria frondosa* inhibits OS adhesion and migration by regulating cytoskeleton remodelling via the Rac1/PAK1/LIMK1/cofilin signalling axis [158].

Regarding LIMK2, several works have shown the crucial role of the RhoA-ROCK-LIMK2 cascade in OS development, particularly in the ability of OS cells to migrate. Wang and Coll demonstrated that BMPR2 functions as a pro-metastatic oncogene in vitro and in vivo. Indeed, BMPR2 silencing in 143B OS cells reduces cell invasion and lamellipodia extension and their metastatic potential in vivo. Mechanistically, the authors demonstrated that LIMK2 is phosphorylated and activated by BMPR2 through the RhoA/ROCK pathway [159]. In this context, Ren and Coll showed that PD-L2 knockdown inactivates this signalling via BMPR2. Hence, PD-L2 knockdown attenuated migration and invasion of OS cells by inactivating RhoA/ROCK/LIMK2 signalling, suppressing epithelial-mesenchymal transition (EMT), and inhibiting autophagy [160]. The role of the PDL2/RhoA/LIMK2 cascade in OS development has also been demonstrated in the ability of VEGFR to stimulate OS metastasis. Zheng and Coll demonstrated that a chemical antagonist of VEGFR or VEGFR2 knockdown downregulates the STAT3 and RhoA-ROCK-LIMK2 pathways in HOS and U2OS cell lines, attenuating OS cell migration and invasion [161]. This RhoA/LIMK2 cascade appears to be equally important in allowing EGF to stimulate OS cell migration. Accordingly, Wang and Coll showed that ROCK, LIMK2, and cofilin, downstream of Rho A, are activated by EGF in MG63 OS cells, which results in actin fibre formation and cell migration. In addition, by chemical loss-of-function inhibitors, the inhibition of LIMK2 prevents actin fibre formation and cell migration in MG63 cells [162].

## 4. Conclusions

Thus, the cumulative evidence demonstrates the relevance of targeting LIMKs in OS. Indeed, targeting LIMK1 could inhibit not only the growth of primary tumours by acting on the control of OS cell proliferation but also the metastatic development by acting on the capacity of OS cells to migrate and invade. Some mechanisms of action of these LIMKs have been identified in OS cells. In particular, the LIMK/cofilin pathway appears to pivotally regulate the dynamization of active filaments. Furthermore, it appears that these LIMKs play a crucial role in the ability of certain mediators, such as BMP or VEGF, to modulate OS cell migration. In addition to metastatic development, one of the major problems in the treatment of OS is its resistance to chemotherapy, which prevents complete remission in some patients. In this context, it is interesting to note that LIMK1 participates in the response of OS cells to chemotherapies such as vincristine.

## Figures and Tables

**Figure 2 cells-10-03542-f002:**
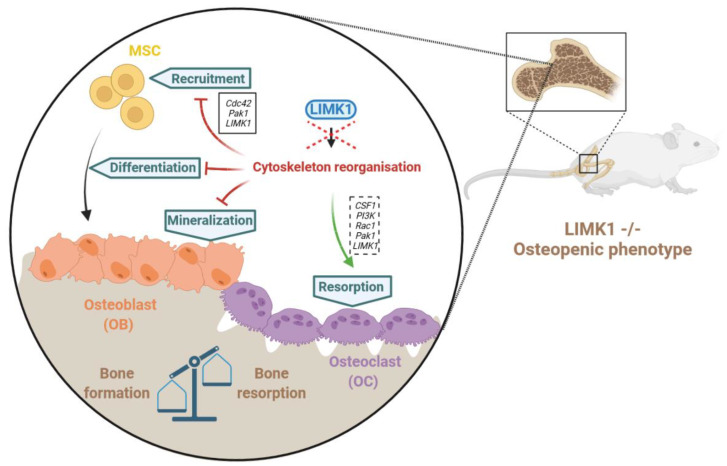
LIMKs and bone remodelling. Bone remodelling depends on a balance between bone formation and degradation. Osteoblasts, which derive from mesenchymal stem cells (MSC), participate in bone formation, while osteoclasts degrade bone. In physiological conditions, there is a balance between bone formation and bone degradation. In LIMK1^−/−^ mice, a disruption of this balance toward bone degradation has been observed. Black plain and discontinued outlines represent the confirmed or putative pathway, respectively.

**Figure 3 cells-10-03542-f003:**
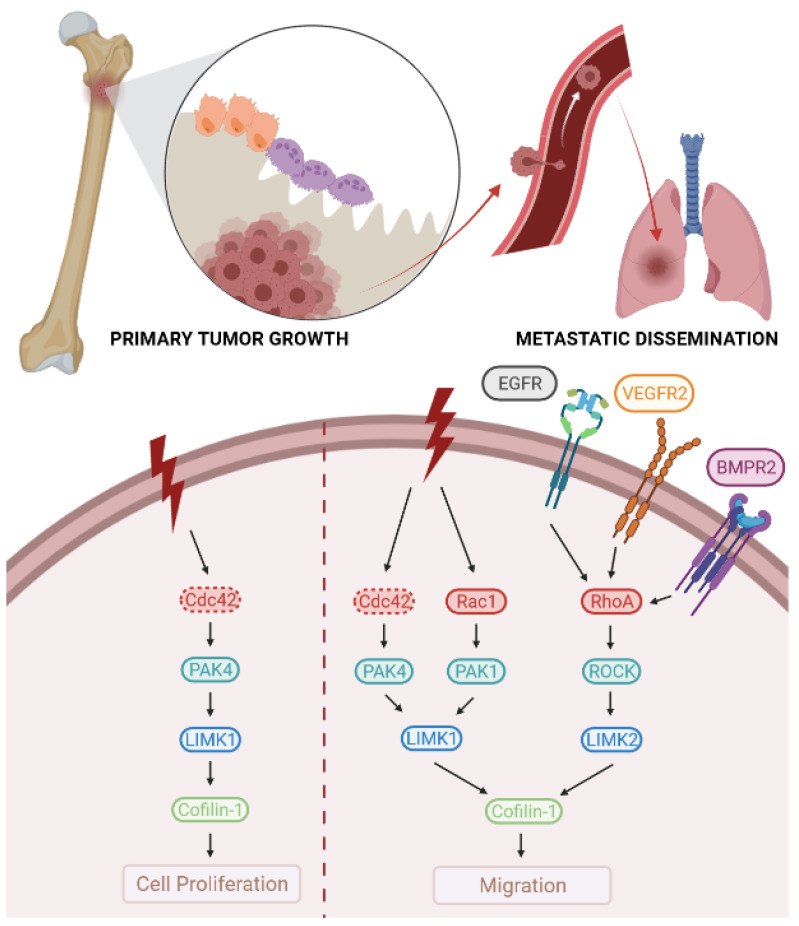
LIMKs and osteosarcoma. LIMK1 regulates the ability of osteosarcoma cells to proliferate and migrate, and thus participates in the control of primary tumour growth and metastatic development. LIMK2 preferentially regulates the ability of osteosarcoma cells to migrate and thus participates in the control of metastatic development.

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
