# Peer review of "LIM Kinases in Osteosarcoma Development"

_cells, 2021, doi:10.3390/cells10123542_

Round 1

Reviewer 1 Report

In the manuscript by Brion et al. titled, "LIM kinases in osteosarcoma development: new therapeutic targets?", the authors comprehensively review the structure and function of LIM kinases 1 and 2 and their putative roles in osteosarcoma. The review of literature supports future studies to explore LIM kinase inhibition to treat aggressive osteosarcoma. The manuscript is well organized and written coherently; however, there are some corrections that need to be applied before the manuscript can be considered for public dissemination. 

Major concerns: 

1) In the abstract the authors state that bone cancers remain a leading cause of death worldwide. This statement as is, is not even remotely accurate. Please consider revising by adding the qualifier among pediatric cancer patients. 

2) The authors do a fantastic job citing previous studies, but they seemed to omit their references for the introduction and halfway through the next paragraph. Please add references. 

3) On line 45, page 2, the authors state that LIMK1-2 encodes a 61-base pair truncated protein corresponding to the catalytic domain. It is confusing in that the reader does not understand is it reduced to 61 base pairs or just 61 base pairs are removed. This is also an example where there was no reference to cross validate the claim. 

Minor changes: 

1) Line 66, page 2 change "seems" to "seem"

2) Table 1, add (metastatic) next to 20/25%

3) Line 370, page 9 change "wich is know" to "which is known"

Author Response

Dear reviewer,

We appreciate the positive feedback on our work " The review of literature supports future studies to explore LIM kinase inhibition to treat aggressive osteosarcoma. The manuscript is well organized and written coherently.”

Major concerns:

1) In the abstract, the sentence concerning the impact of bone cancer in children has been modified according to reviewer's recommendations. New sentence "In this context, despite research and clinical advances during the past decades, bone cancers remain a leading cause of death worldwide among pediatric cancer patients ».

2) We thank the reviewer for their glowing comment "The authors do a fantastic job of citing previous studies."

Regarding works of our laboratory, we have not yet published on the role of LIMK in osteosarcoma but on other signaling pathways such as the hippo pathway which is closely associated functionally with actin filament remodeling and the RhoA/LIMK/cofilin pathway. In this context, some references and some sentences have been added in the paragraph « 3.2.1. Osteosarcoma (OS) ». New sentences « The development of OS is closely associated with the activation of various signaling pathways. Recently, we have shown that the Hippo/YAP pathway is involved in the regulation of OS development both by promoting primary tumor growth through the association of YAP with the transcription factor TEAD and by promoting metastatic development through the association of YAP with the transcription factor Smad3 (Morice et al., Cancers 2020a, Cancers 2020b, Front Oncol 2021). Interestingly, cytoplasmic YAP is able to regulates the expression of Rho GTPase activating protein known as the regulator of the RhoA/LIMK/cofilin pathway (Das et al., J Biol Chem 2016 ; Qiao et al., Cell Reports 2017), suggesting a potential role of the RhoA/LIMK/cofilin cascade in OS developement.»

3) On line 45, page 2, the sentence has been modified for a better understanding. Three references have been added. New sentence: "Two splice variants of LIMK1 are known to date: LIMK1-1 and LIMK1-2. LIMK1-1 is thought to encode the full-length protein, and LIMK1-2, that lacks the kinase domain. (Edwards and Gill, J Biol Chem 1999; Stanyon and Bernard, The International Journal of Biochemistry & Cell Biology, 1999 ; Bernard, Int J Biochem Cell Biol, 2007) ».

Minor changes:

1) Line 66, page 2, "seems" has been replaced with "seem".

2) Table 1, "metastatic" was added after "20/25%"

3) Line 370, page 9 "wich is known" has been changed to "which is known".

We thank the reviewer who allowed us to improve our manuscript. We hope that all the changes made to our original manuscript satisfactorily address the reviewer’s issues and that this revised version is now acceptable for publication.

Sincerely

Franck Verrecchia

Reviewer 2 Report

This is an interesting and well written review.

I have the following three concerns:

L.29-30: Percentages' meanings are not clear and not referenced.

"schematically" is used seven times and it is too much. Some points have not to é"schematically" descriebed but to the contrary, in details.

I would suggest to modify the title sin ce it doesn't fit very well with the main text. Actually dat presented about LMIK are sometimes overstated:

L.367: "LIMK1 also seem to controlOS cell migration" but in the referred paper it's the silencing of LIMK1 in vincristine-resistant cells that decreases their ability to migrate.

So, I am not sure that the question about LIMKs as  new therapeutic targets, as mentionned in the title, would be the good one.

Author Response

Dear reviewer,

We appreciate the positive feedback on our work " This is an interesting and well written review.”

Regarding the various comments and recommendations

  • 29-30: Percentages' meanings are not clear and not referenced.

This paragraph has been rewritten for a better understanding and references have been added. New sentences “While the five-year survival rate is greater than 78% in the absence of metastases at diagnosis or for good responders, this survival rate drops to 25% for metastatic or recurrent OS (Gaspar et al., Eur J Cancer 2018; Heare et al., Curr Opin Pediatr 2009; Moore and Luu, Cancer Treat Res. 2014; Kempf-Bielacket al., J Clin Oncol. 23 2005)”

  • "schematically" is used seven times and it is too much

We agree with these remarks. These "language" defaults have been corrected and precision has been added to the paragraph. “2.2. LIMKs and cancer”: Additional sentence “Thus, among the many activators of LIMK, members of the Rho family of small GTPases (Rho, Rac and Cdc42) and their effectors (ROCK, PAK and MK2) are implicated in the progression of various human cancers to invasive and met-astatic stages (Manetti, Curr Cancer Drug Targets, 2012)”.

  • I would suggest to modify the title since it doesn't fit very well with the main text.

We agree with this remark. The title has been simplified to better match the data of the text. New Title " LIM kinases in osteosarcoma development”.

We thank the reviewer who allowed us to improve our manuscript. We hope that all the changes made to our original manuscript satisfactorily address the reviewer’s issues and that this revised version is now acceptable for publication.

Sincerely

Franck Verrecchia

Round 2

Reviewer 1 Report

The authors have adequately addressed my previous concerns.